# Recovery time of a plasma-wakefield accelerator

R. D'Arcy[1✉], J. Chappell[2], J. Beinortaite[1,2], S. Diederichs[1,3], G. Boyle[1], B. Foster[4], M. J. Garland[1], P. Gonzalez Caminal[1,3], C. A. Lindstrøm[1], G. Loisch[1], S. Schreiber[1], S. Schröder[1], R. J. Shalloo[1], M. Thévenet[1], S. Wesch[1], M. Wing[1,2] & J. Osterhoff[1]

The interaction of intense particle bunches with plasma can give rise to plasma wakes[1,2] capable of sustaining gigavolt-per-metre electric fields[3,4], which are orders of magnitude higher than provided by state-of-the-art radio-frequency technology[5]. Plasma wakefields can, therefore, strongly accelerate charged particles and offer the opportunity to reach higher particle energies with smaller and hence more widely available accelerator facilities. However, the luminosity and brilliance demands of high-energy physics and photon science require particle bunches to be accelerated at repetition rates of thousands or even millions per second, which are orders of magnitude higher than demonstrated with plasma-wakefield technology[6,7]. Here we investigate the upper limit on repetition rates of beam-driven plasma accelerators by measuring the time it takes for the plasma to recover to its initial state after perturbation by a wakefield. The many-nanosecond-level recovery time measured establishes the in-principle attainability of megahertz rates of acceleration in plasmas. The experimental signatures of the perturbation are well described by simulations of a temporally evolving parabolic ion channel, transferring energy from the collapsing wake to the surrounding media. This result establishes that plasma-wakefield modules could be developed as feasible high-repetition-rate energy boosters at current and future particle-physics and photon-science facilities.

Radio-frequency (RF) accelerator technology has driven material-science and particle-physics research for the past century. Maximizing accelerated charge per second maximizes the brilliance of free-electron lasers[8,9] and the luminosity of future linear colliders[10]. The high quality factor of metallic super-conducting RF cavities, maintaining up to $10^{11}$ electromagnetic oscillations before substantial dissipation[11], enables the efficient extraction of energy over long bunch trains at high repetition rates. However, because of electrical breakdowns, such cavities are incapable of sustaining electric fields greater than about 40 megavolts-per-metre (MV m$^{-1}$)[5], thus necessitating large facilities to reach high particle energies.

Plasma wakes[1,2] driven by intense particle bunches[3,4] represent a disruptive development owing to their ability to sustain gigavolt-per-metre (GV m$^{-1}$) electric fields. To exceed the luminosity and brilliance demands of current facilities, however, plasma-acceleration techniques must also be capable of accelerating particle bunches at high repetition rates. The electromagnetic fields in plasma, unlike those in RF cavities, are significantly damped after only a few oscillations, at which point the plasma wake collapses and its stored energy dissipates into the surrounding media. Therefore, it is essential to use the first oscillation of the wakefield for acceleration and then wait for the perturbed background plasma to recover to approximately its initial state before the next acceleration. This recovery time places an upper limit on the maximum achievable repetition rate of plasma accelerators.

During excitation of a particle-beam-driven plasma wake, electrons and ions are separated by the space-charge field of the intense, relativistic particle beam. On the timescale of this motion, defined by the plasma-electron frequency, the plasma ions are typically treated as being stationary. On longer timescales, however, the plasma ions move towards the beam axis[12–15], impelled there by the strong radial electric fields of the beam and the decaying plasma wave. The timescale of this movement is defined by the mass and charge of the ions and the strength of the radial fields. After the on-axis ion density reaches a maximum, the ion wave continues to propagate outwards, assisted by the pressure gradient between regions of differing plasma density and temperature, until uniform conditions are re-established. Laser-based optical-probing techniques have been used to observe the onset of ion motion[16] and subsequent ion acoustic waves[17] in the form of on-axis electron-density peaks. To measure the collective ion motion, a diagnostic technique based on the interaction of multiple electron bunches with plasma has been developed.

The collective motion of ions was measured at the plasma-wakefield research facility FLASHForward[18]. The initial plasma, with electron density $1.75 \times 10^{16}$ cm$^{-3}$, was generated by sending a high-voltage discharge through a capillary filled with argon gas (Methods). A large-amplitude, non-linear wakefield (1.6 GV m$^{-1}$) was then driven by an intense leading electron bunch (1.5 kA peak current), produced by a photocathode laser temporally locked to the plasma generation, and accelerated by

[1]Deutsches Elektronen-Synchrotron DESY, Hamburg, Germany. [2]University College London, London, UK. [3]Universität Hamburg, Hamburg, Germany. [4]John Adams Institute, Department of Physics, University of Oxford, Oxford, UK. ✉e-mail: richard.darcy@desy.de

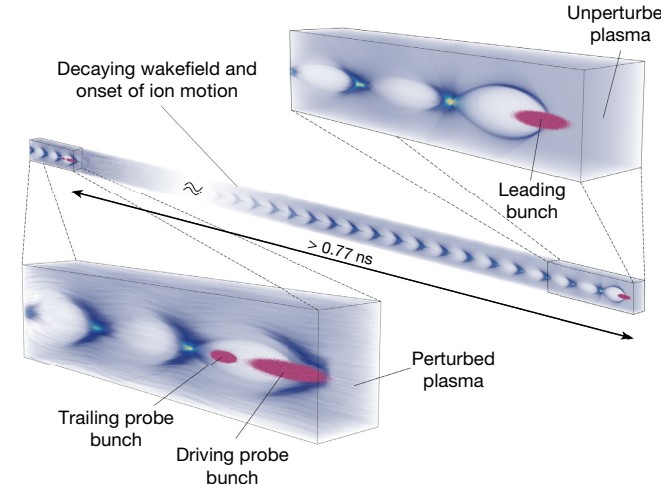

Fig. 1 | Conceptual representation of the plasma probe process. For the perturbed measurements, a leading bunch drives a wakefield, which in turn stimulates motion of the plasma ions. The two probe bunches sample the perturbed plasma in increments of 0.77 ns after the temporally locked leading bunch. For the unperturbed measurements, the procedure is the same but without the presence of the leading bunch. The rendering was performed using VisualPIC[33].

the FLASH linac to 1,061 MeV (ref. [19]). A probe bunch, with different parameters to the leading bunch, was produced by a second photo-cathode laser and placed in a later RF bucket (Methods). The probe bunch (accelerated to 1,054 MeV) was bisected into a pair of bunches in the FLASHForward experimental beamline—the first 'driving probe' bunch driving a subsequent non-linear wakefield and the second 'trailing probe' bunch travelling behind in its wake. The two probe bunches propagated through the perturbed plasma at varying times after the leading bunch (Fig. 1), thereby driving a second plasma wake the properties of which depend on the state of the perturbed plasma. By analysing the two probe bunches, the perturbed plasma can be sampled with temporal resolution defined by the 1.3 GHz frequency of the RF accelerating cavities in the FLASH linac. The recovery time of the plasma is defined as the point at which the properties of the probe bunches are consistent with those measured after interaction with an unperturbed plasma, that is, in the absence of the leading bunch.

The results of a scan varying the separation of the leading and probe bunches can be seen in Fig. 2, which was generated by dispersing all three bunches in a dipole magnet, focusing them on a scintillating screen after the interaction and subtracting the overlapping energy spectra, averaged over many bunches, of the leading bunch from that of the driving probe bunch (Methods). In the case of the unperturbed plasma, both the energy spectra and transverse distributions (Fig. 2a) remain approximately constant over the duration of the 160 ns scan (see Extended Data Fig. 1 for separations 70–160 ns), with gradual changes due to the slow dynamic evolution of the background plasma as it undergoes recombination and is gradually expelled from the open capillary ends[20]. However, in the case of the perturbed plasma, the energy spectra and transverse distributions (Fig. 2b) vary significantly over the same timescale until approximately 63 ns, at which point all residuals are compatible with zero (Fig. 2c), indicating that the probe-bunch properties are consistent with those of the unperturbed case. That this is the case over longer timescales, up to 160 ns, can be seen in Fig. 2c and Extended Data Fig. 1. The recovery time for this operational state translates to an interbunch repetition-rate upper limit of $O$(10 MHz).

The dominant physics mechanisms and timescales defining the ion motion can be understood from the features of Fig. 2. The timescale over which an on-axis ion-density peak is generated depends on the longitudinally integrated radial fields of the electron bunch and plasma

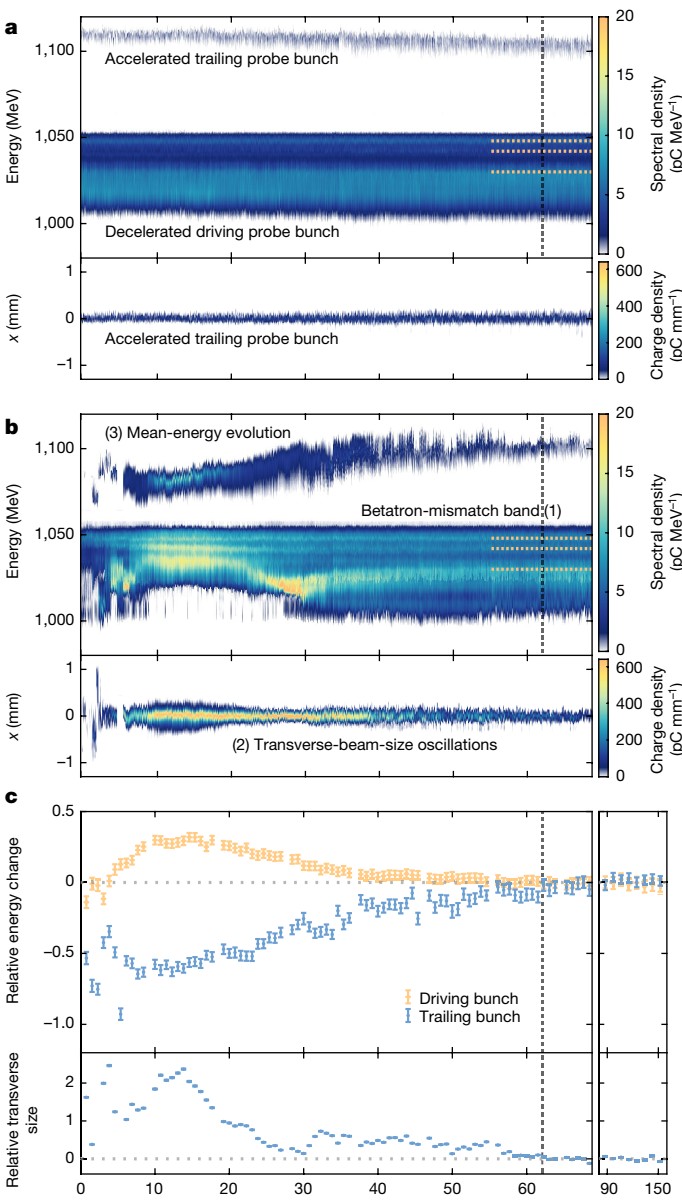

Fig. 2 | Recovery time of a beam-driven plasma wake. a, The energy spectra and transverse distributions of the probe bunches after interaction with an unperturbed plasma. b, The same as in a but after interaction with a plasma perturbed by the leading bunch. Imperfections in the procedure used to subtract the overlapping spectra of the leading bunch from the driving probe bunch (Methods) lead to small systematic differences between the energy spectra of a and b at late timescales, for example, in the large betatron-mismatch band at approximately 1,030 MeV. Larger trailing-probe-bunch charge is also seen in b at shorter timescales due to higher coupling between the plasma and downstream capturing optics. c, The residuals (Methods) between the energy spectra and transverse bunch size of both the unperturbed and perturbed datasets. Extended data up to 160 ns is shown on a compressed horizontal timescale in the right-hand panel. The error bar represents the standard error of the mean. The recovery time, indicated by the black dashed vertical line, is reached when all three residuals are consistent with zero. The three experimental signatures of ion motion are enumerated in b, with orange dashed bands for the first signature added to a and b to guide the eye.

wave. Considering only the fields generated by the leading electron bunch, it is estimated that the on-axis ion density for a singly ionized argon plasma will reach its maximum at $0.5 \pm 0.2$ ns (Methods)—an upper-bound estimate yet still outside the 0.77 ns temporal resolution

of the diagnostic. The ensuing expansion of the on-axis spike manifests itself as an outwardly propagating ion acoustic wave[21]. As this wave propagates outwards, the on-axis ion density is expected to fall and be subsequently replenished by inwardly streaming cold plasma, essentially unperturbed by the initial wakefield. The speed of each counter-propagating wave is proportional to $\sqrt{T_e/m_i}$, where $T_e$ is the plasma-electron temperature and $m_i$ is the plasma-ion mass[21,22]. On the basis of the results of previous investigations[12–17], we describe the temporally evolving non-uniformity of the radial density profile to lowest order (that is, parabolic) with the form $n(r) = n_0(1 + \alpha r^2)$, where the on-axis ion density $n_0$ and curvature of the channel $\alpha$ are time dependent and $r$ represents the radial distance from the axis.

Ideally, self-consistent particle-in-cell (PIC) simulations of a wakefield driven by the leading bunch, followed by the evolution of the plasma over tens of nanoseconds, would be carried out to understand the details of Fig. 2. However, such simulations are out of reach using current numerical methods owing to prohibitively high computational demands and the accumulation of debilitating numerical noise[23]. In light of this, our approach is to show that the results can be plausibly attributed to ion motion by investigating the experimental signatures of Fig. 2, using them to quantify $n_0$ and $\alpha$ through the application of non-linear[24,25] and linear[4] plasma-wakefield theory, respectively, and testing the efficacy of the derived profiles by using them as initial conditions in PIC simulations of the probe process over short timescales. The three experimental signatures investigated are labelled in Fig. 2: (1) betatron-mismatch bands of the driving-probe-bunch energy spectra; (2) transverse size of the trailing probe bunch; (3) mean-energy evolution of the two probe bunches.

The first signature is a manifestation of the head-to-tail growth of the focusing force acting on the head of the driver during the build-up of the wakefield, which results in differing betatron-oscillation frequencies for each longitudinal beam slice in that region. The energies at which these bands of raised intensity appear are constant, not only over the slowly decaying density range of Fig. 2 but also over many orders of magnitude beyond (Extended Data Fig. 2), illustrating that they are independent of the on-axis density. This is due to the low-density head of the driving probe bunch producing a linear wakefield response in that region: for the slice experiencing a given deceleration, the betatron wavelength at the position of this slice does not depend on the density (Methods). In the perturbed case, the energy of these bands shifts as a function of separation, and this shift occurs most significantly at shorter bunch separations. This is a result of the non-linear focusing force generated by the parabolic ion density either increasing or decreasing off axis. The deviation in energy of these bands from the unperturbed values may be used to determine the $\alpha$ parameter of the parabola at each separation (Methods).

The second signature relates to the transverse size of the trailing probe bunch. As implied by the trailing-probe-bunch energy spectra (Fig. 2a), the unperturbed on-axis plasma density gradually decreases over time. The longitudinal-wakefield amplitude experienced by the trailing probe bunch correspondingly decreases and its betatron-oscillation frequency is modified. It therefore accrues a betatron phase over the plasma length that varies with $n_0$, resulting in a differing divergence and transverse size at the plasma exit. In the perturbed case of Fig. 2b, the change in transverse beam size is much more pronounced than the unperturbed case, indicating a significant change of $n_0$ during this time. As such, the transverse beam size provides an experimental signature for $n_0$ that is independent of $\alpha$, because the transverse size of the trailing bunch is small compared with that of the parabolic channel (Methods).

Figure 3 shows experimental results obtained by operating with beam conditions designed to amplify the first two experimental signatures: $\alpha$ is determined by increasing the beam size of the driving-probe-bunch head at the plasma entrance, such that it samples a larger transverse range of the parabolic ion profile while still remaining small

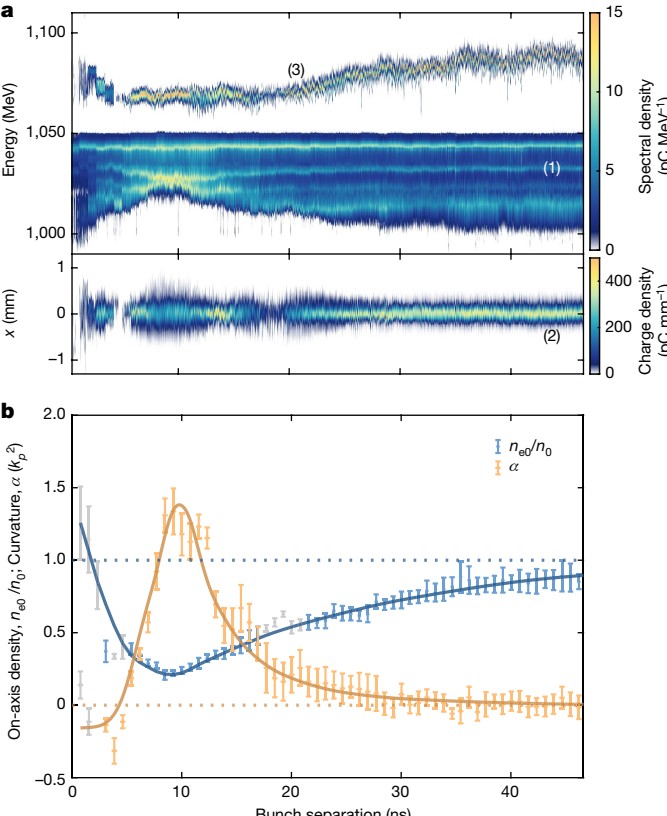

**Fig. 3 | Derivation of the parabolic ion channel properties. a**, Energy spectra and transverse distributions of the driving and trailing probe bunches after interaction with the same perturbed plasma sampled in Fig. 2b. Differences between the spectra and transverse distributions of Fig. 2b arise from modification to the probe bunches made to enhance the strength of the experimental signatures. **b**, Curvature and on-axis ion-density values derived from the first two experimental signatures (Methods) highlighted in **a**. The line fits were calculated with a cubic spline. The error bars reflect the experimental uncertainties, which are dominated by the fits to measurements of the relative trailing beam size and the position of the focal lines. Values derived from separations with significant charge loss (>50%), which therefore carry an additional systematic uncertainty beyond the fitting errors shown, are highlighted in grey. This primarily occurs (1) at short timescales for both probe bunches due to their interaction with the decreasing off-axis focusing forces of the on-axis density spike and (2) at approximately 20 ns for the highly divergent trailing probe bunch owing to clipping in the capturing quadrupoles downstream of the plasma capillary.

relative to the parabolic channel width; $n_0$ is measured by reducing the length of the trailing probe bunch, such that it samples a smaller longitudinal-wakefield phase range and, therefore, the oscillations remain mostly coherent. Figure 3 shows that the magnitude of both the betatron-mismatch-band shift and trailing-probe-bunch transverse oscillations have increased compared with Fig. 2, while the third experimental signature—the large relative change in the mean energies of the driving and trailing probe bunches—has been maintained. This third signature arises from a combination of the variable $n_0$ and $\alpha$, which modifies both the extent and electric field of the plasma cavity. In the case of the largest energy change at approximately 10 ns separation, $n_0$ and $\alpha$ combine to lengthen the wakefield cavity and reduce the electric-field strengths observed by both probe bunches compared with the unperturbed case. This drop in the accelerating field reduces the mean energy of the trailing bunch from approximately 1.10 GeV to approximately 1.07 GeV.

The values of $n_0$ and $\alpha$ with associated uncertainties—corresponding to the first two experimental signatures of Fig. 3a—are derived

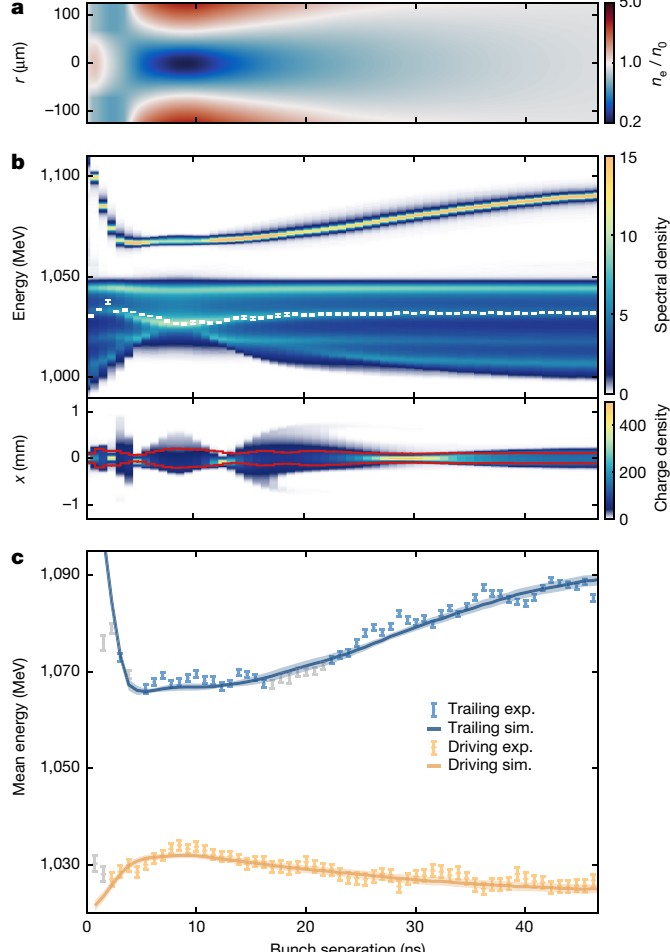

**Fig. 4 | Numerical reconstruction of the experimental results. a**, The radial-ion-density map interpolated from the experimental results of Fig. 3b. **b**, The simulated energy spectra and transverse distributions of the probe bunches after interaction with the radial-ion-density map of **a**. The mean values and uncertainties of the experimental betatron-mismatch band (white error bars) as well as the r.m.s. values of the experimental transverse beam size (red solid lines) are overlaid, demonstrating good agreement with experiment except at approximately 20 ns in the transverse distribution, at which time some of the charge of the highly divergent trailing probe bunch is lost owing to clipping in the capturing quadrupoles downstream of the plasma capillary. **c**, A comparison of the third experimental (exp.) signature of Fig. 3a–not used in the fitting procedures for $n_0$ and $\alpha$–with the equivalent simulated (sim.) values of **b**. Error bars represent the standard error of the mean. As in Fig. 3b, the grey data points highlight separations at which some of the charge was lost.

for each separation by using a fit-function based on non-linear and linear plasma-wakefield theory, respectively (Fig. 3b) (Methods). At the shortest separations, $n_0$ is higher than the unperturbed value and $\alpha$ is negative, both of which indicate that the probe beam is sampling the end of the on-axis peak at short times. Beyond this time the ion channel begins to form, with the on-axis density decreasing to a minimum and the curvature increasing to a maximum by approximately 10 ns. Once the channel reaches this deepest point, the inwardly propagating cold plasma refills the depleted channel, with the on-axis ion density and curvature slowly tending back towards their unperturbed values.

The derived values of Fig. 3b were used to construct an evolving two-dimensional density map (Fig. 4a), used as input for PIC simulations over short timescales to test their validity. For accurate modelling, six-dimensional (6D) distributions of both the driving and trailing bunches were reconstructed from transverse- and longitudinal-phase-space measurements (Methods). The simulation

results are shown in Fig. 4b, with the corresponding experimental signatures overlaid. The third experimental signature, which was not used in the derivation of the parabolic ion channel, is compared with the corresponding simulated values in Fig. 4c. The disagreement at the very earliest times is caused by initial charge loss, as described in Fig. 3b. The excellent agreement for later times provides independent validation of the derivation procedure and the physics model.

The success of the evolving-ion-channel model establishes that ion motion is the key mechanism in understanding the experimental data. Thus, the timescales involved should depend on the ion mass and plasma temperature[21,22]. Argon was selected for this study to match the temporal range of the diagnostic. However, by operating with, for example, hydrogen, the recovery time should be reduced by a factor of $\sqrt{m_{Ar}/m_H} \approx 6$, that is, to approximately 10 ns. The timescales of the on-axis peak and refilling of the channel are also expected to depend on the wakefield strength and initial plasma-electron density. A detailed parametric assessment of these dependencies will be a topic of future study.

The results reported here place an upper limit on the interbunch repetition rate for plasma-wakefield accelerators arising from fundamental plasma-physics processes. To operate a future plasma-based facility at high repetition rates, however, other factors must also be taken into account. One such example is cumulative heating of the plasma from the acceleration of long bunch trains. PIC simulations explored elsewhere[26–28] suggest that thermal effects of $O$(100 eV) are not expected to modify significantly the acceleration process of the accelerating bunch but temperature effects beyond this range remain unexplored. Another important consideration is the macroscopic effect of this heat deposition, which could lead to damage of the plasma source if left untreated. Any possible effects of plasma heating, both on the plasma-wakefield properties and the structural integrity of the plasma cell, may be mitigated by running with bunch trains, as indeed is proposed for both the ILC[29] and CLIC[30] accelerators. Each bunch train would contain many bunches separated by the recovery time (at minimum), with each bunch train separated by a time sufficient to enable any heating effects to be reduced to an acceptable level. Furthermore, plasma sources designed specifically with high repetition rate in mind[31,32] may be utilized, for which the cooling dynamics would be entirely different to the source type used here but the motion of the ions, as characterized in this work, would be unchanged. The importance of the results reported here is to establish that fundamental plasma-physics processes central to the operation of plasma accelerators do not preclude their application to current and future high-repetition-rate facilities.

In summary, we have measured the recovery time of a GV m$^{-1}$ gradient beam-driven plasma accelerator to be 63 ns in an argon plasma of density $1.75 \times 10^{16}$ cm$^{-3}$. Simulations confirm that the data can be explained by the long-term evolution of a parabolic ion profile produced by the transfer of energy in the system after the plasma wake breaks down. The return of the perturbed plasma to the unperturbed state in such a timescale establishes that megahertz interbunch repetition rates are supported and hence luminosities and brilliances beyond the state of the art are in principle attainable in plasma-wakefield-accelerator facilities of the future.

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

## Methods

### Plasma generation and characterization

A high-voltage discharge was used to create the plasma, ignited by a thyratron switch operating at a breakdown voltage of 25 kV, supplying approximately 500 A for a duration of 400 ns. The plasma was contained within a 1.5-mm-diameter, 50-mm-long capillary milled from two slabs of sapphire, mounted in a PEEK plastic holder, all mounted on a hexapod platform for high-precision alignment. A continuous flow of argon was supplied through two internal gas inlets from a buffer volume at a 10 mbar backing pressure. The gas escaped the open-ended capillary through holed copper electrodes (cathode upstream, anode downstream) into a large 500-mm-diameter vacuum chamber pumped to an ambient pressure of $4.3 \times 10^{-3}$ mbar. Broadening of spectral lines[34] enabled the density at the longitudinal centre of the plasma cell to be resolved[35]. The profile and evolution of the plasma density were recorded from the start of the discharge (0 µs) and to just after the arrival time of the electron beam (2.6 µs after discharge). The argon was doped with 3% hydrogen (defined by atomic density) to spectrally broaden the H-alpha line. The density measurements were fitted to obtain the plasma density, $(1.75 \pm 0.27) \times 10^{16}$ cm$^{-3}$, at the arrival time of the electron beam[20].

### Electron-bunch generation and transport

The leading and probe electron bunches were generated by two distinct photocathode lasers. The maximum repetition rate of the two individual photocathode lasers[36] is 1 and 3 MHz, which is defined by the fastest rate at which the Pockels-cell drivers can pick pulses from the laser oscillator. However, the base low-level frequency of the FLASH facility is 1.3 GHz. The 3 MHz limit of a single photocathode laser can, therefore, be overcome by placing separate photocathode-laser pulses in consecutive 1.3 GHz RF buckets. This defines the 0.77 ns (1/1.3 GHz) resolution of the perturbation diagnostic. The timing between the leading bunch and the probe bunch can be increased by incrementally shifting to later RF buckets in 0.77 ns steps. The two lasers produce pulses of differing root mean square (r.m.s.) length, 4 and 6 ps, which translates directly into electron bunches of differing length. To compensate for variable space-charge effects in the early stages of the FLASH superconducting linac arising from electron bunches of different length but equal charge, the electron bunch charges were scaled accordingly to be 700 pC (leading) and 900 pC (probe). These two bunches were then accelerated to a mean particle energy of 1,061 MeV and 1,054 MeV, respectively. The bunches were compressed in two magnetic chicanes. A kicker magnet was used to extract the bunches into the dispersive section of the FLASHForward beamline, where a set of three collimators were used to manipulate the bunch-current profile—one wedge to bisect the probe bunch and two outer blocks to remove high- and low-energy electrons[37]. The exact positioning of the three wedges was varied slightly between the two working points of Fig. 2 (WP1) and Fig. 3 (WP2) to accentuate certain experimental signals. For both working points, the remaining charge in the leading bunch was constant at 590 pC. For WP1, the probe bunch charges after scraping were 320 pC (driving) and 90 pC (trailing). For WP2, they were 242 pC (driving) and 48 pC (trailing). In addition, the final-focussing quadrupoles were modified to enlarge the head of the driving beam at the plasma entrance. This also reduced the density of the head of the leading bunch such that the strength of the wakefield it generated was correspondingly reduced by 3% (as measured by the maximum energy loss of the leading bunch in the electron spectrometer). As the probe bunch had a linear correlation in longitudinal phase space, its length could be reduced to reach that desired for WP2 by cutting away energy slices from the rear of the bunch. Toroids were used to measure the bunch charge before and after the energy collimation. A set of quadrupoles was used to tightly focus the beam at the location of the plasma cell. These matching quadrupoles were set to focus the beam to a waist close to the plasma entrance. The waist location and beta function were then measured and fine-tuned with a precision of $O(10 \text{ mm})$ using a new jitter-based measurement technique[38]. The same two cavity-based beam-position monitors (50 cm upstream and 50 cm downstream of the plasma) were used for beam alignment. Five differential pumping stations enabled a windowless vacuum-to-plasma transition, ensuring high beam quality while also meeting the ultrahigh vacuum requirements of the superconducting FLASH accelerator.

### Electron imaging spectrometer

A dipole magnet was used to perform energy dispersion of the beam vertically onto a LANEX (fine) screen mounted just outside the 1-mm-thick stainless-steel vacuum chamber wall, approximately 3 m downstream of the plasma cell. Five quadrupoles (acting as a triplet) located just upstream of the dipole were used to point-to-point image the beam from the plasma-cell exit (the object plane) to the screen (the image plane) with a magnification of $R_{11} = -5$ (horizontally) and $R_{33} = -0.97$ (vertically), where $R$ is the object-to-image-plane transfer matrix. The spatial resolution of the optical system was approximately 50 µm (that is, approximately 2 pixels), corresponding to an energy resolution of 0.05% for particles close to the imaged energy. Away from this imaged energy, the energy resolution degrades depending on the vertical divergence of the bunch. The recorded two-dimensional images in the $(x, E)$ plane may be collapsed onto a single axis to produce a spectral density map in either $x$ or $E$. The stacking of these maps—in this case a function of bunch separation—is displayed as a waterfall plot in Figs. 2a,b, 3a and 4b.

### Spectrometer image subtraction

In these measurements, multiple bunches interact with the electron-spectrometer scintillating screen in its scintillation lifetime (measured to be approximately 380 µs), leading to overlapping signals in both space and time. A subtraction technique was developed[39] to enable reconstruction of the spectra of the probe bunch. This technique uses $O(100)$ measurements of only the leading bunch to predict its scintillation signal (based on its charge) and remove this from the spectrometer images in the case of the perturbed plasma. This subtraction process contributes to the systematic uncertainty included in calculations of the energy and transverse distributions of the probe bunch and is of the order of 10%; the magnitude of the systematic uncertainty is calculated by pixel-by-pixel comparisons of the measured scintillation signal of the leading bunch only and its corresponding predicted signal for each of the $O(100)$ events. Imperfections in this subtraction procedure lead to small differences in the driving-probe-bunch energy spectra (Fig. 2a versus Fig. 2b). The three-bunch setup used here (a single leading bunch followed by two probe bunches) means the scintillation signal from the trailing probe bunch is unaffected by the subtraction procedure (as there is no overlap of the trailing probe bunch with any other bunch on the scintillation screen) and hence its properties provide the cleanest signal, motivating its use to define the relaxation of the perturbation. All properties of the trailing bunches are compared with optics set to image an energy of 1,100 MeV to improve the resolution of the trailing probe bunch. However, comparisons between the mean energies of the driving probe bunch in the perturbed and unperturbed cases (orange data points in Fig. 2c) are performed with a spectrometer imaging energy of 1,050 MeV. In this case, the subtraction technique is more accurate (with a few per cent systematic uncertainty) as the change in imaging energy minimizes imaging errors in the driving-probe-bunch spectra.

### Definition of residuals

Three separate residuals are used to define the convergence of the perturbed plasma to the unperturbed state. The first two correspond to measurements of the change in mean energy of the driving and trailing probe bunches. This is referred to as the 'relative energy change' in Fig. 2c and is calculated from

$$\frac{\mu_{E,\mathrm{u}} - \mu_{E,\mathrm{p}}}{\Delta\mu_E},$$

where $\mu_{E,\mathrm{u/p}}$ represents the mean energy of the unperturbed (u) or perturbed (p) bunch and $\Delta\mu_E$ represents the average energy gain and loss of the trailing and driving probe bunch, respectively, in the unperturbed scheme relative to the energy of that bunch without plasma interaction. The third residual is the 'relative transverse bunch size', calculated from

$$\frac{\sigma_{x,\mathrm{p}} - \sigma_{x,\mathrm{u}}}{\sigma_{x,\mathrm{u}}},$$

where $\sigma_{x,\mathrm{u/p}}$ represents the transverse size of the trailing probe bunch in the unperturbed (u) or perturbed (p) scheme measured in the plane of the electron spectrometer. The bunch separation beyond which all three residuals return to, and remain at, zero within experimental uncertainties is defined as the recovery time of the plasma.

## Timescale for the formation of an on-axis density spike

In a plasma-wakefield accelerator, ions inside the plasma wake focus the electrons in the passing beam. In the process, the beam electrons will also exert an equal but opposite force on the ions, which varies both in time, $t$, and in space, $r$. Assuming, for simplicity, a cylindrical bunch of area $2\pi\sigma_r^2$ and a current profile $I(t)$, the radial force on the ions in the radius of the beam is

$$F_r(t,r) = \frac{eZ_i I(t)r}{4\pi c\sigma_r^2\varepsilon_0},$$

where $Z_i$ is the ionization state of the ions, and $e$, $c$ and $\varepsilon_0$ are the electron charge, speed of light in a vacuum and permittivity, respectively. Rosenzweig et al.[14] used a similar starting point to model the motion of ions within the initial plasma cavity. However, in the present study the motion of the ions is negligible on the timescale of the plasma-electron frequency. Instead, the total radial impulse,

$$\Delta p_r(r) = \int F_r(t,r)\,\mathrm{d}t = \frac{eZ_i Q r}{4\pi c\sigma_r^2\varepsilon_0},$$

induces a (non-relativistic) radial ion velocity $\Delta v_r = \Delta p_r(r)/m_i$, where $m_i$ is the ion mass and $Q = \int I(t)\,\mathrm{d}t$ is the total bunch charge. Assuming that plasma electrons do not significantly alter the collective ion motion, the ions are all 'focused' onto the axis in a time

$$\Delta t_{\mathrm{spike}} = \frac{r}{v_r} = \frac{4\pi c\varepsilon_0 m_i\sigma_r^2}{eZ_i Q},$$

which represents an approximate upper bound to the timescale of the on-axis ion-peak generation. For this experiment (Fig. 2), operating in singly ionized argon ($Z_i = 1$, $m_i = 6.64\times10^{-26}\,\mathrm{kg}$) with an average leading-bunch charge of 590 pC and r.m.s. transverse beam size of $5\pm1\,\mu\mathrm{m}$, the resulting formation time for the density spike is estimated to be approximately $0.5\pm0.2$ ns.

## Origin of the density-independent betatron-mismatch bands

The driving probe bunch occupies a large range of wakefield phase, and hence longitudinal-field amplitude, for the range of plasma-electron densities used in the experiments. As a result, the betatron phase advance, and, therefore, the divergence of individual energy slices, varies significantly across the bunch at the plasma exit. When fixing the focal energy of the capturing optics, this variation manifests itself as bands of raised intensity at the spectrometer screen, separated by an $n\pi$ phase advance. In the unperturbed case, the energy at which these

bands appear is constant over an orders-of-magnitude plasma-density range (Extended Data Fig. 2). This is due to the linear wakefield response generated by the head of the driving probe bunch, which is relatively low in density because of the coherent synchrotron radiation induced during the transport of the bunch to the plasma.

In this regime, the focusing and decelerating fields at a given longitudinal slice are linked with the focusing-field strength[40,41], which is approximated as

$$\frac{F_r}{r} = -\left(\frac{8\varepsilon_0 L\Delta^3}{9e^2 n_{\mathrm{b0}}w^2}\right)^{1/2},$$

where $L$ is the length of the beam, $\Delta$ is the magnitude of deceleration of that slice, $n_{\mathrm{b0}}$ is the peak bunch density and $w$ is the width of the beam. In this region of the driving probe bunch, the current profile can be approximated as being longitudinally triangular ($L = 60\,\mu\mathrm{m}$) and transversely Gaussian (r.m.s. $w = 40\,\mu\mathrm{m}$), with a charge of 125 pC (giving $n_{\mathrm{b0}} \approx 1.2\times10^{16}\,\mathrm{cm}^{-3}$). To obtain this expression, one can get the analytic expression for the pseudo-potential in the beam using Green functions[40] and extract the corresponding transverse and focusing forces. The expression can then be readily derived. With these values, the model predicts three shifts of $\pi$ in the final phase of betatron oscillation over the head of the bunch, all separated by approximately 10 MeV, that is, in good agreement with the experimental results of Fig. 3a.

## Quantification of the betatron-mismatch bands

The energy of the main betatron-mismatch band in the perturbed-plasma case (Fig. 3a) is calculated for each separation by fitting a peak to the spectrometer image projected onto the energy-dispersed axis. The mean energy of these bands is given by the peak of the fit, with error bars representing the average full-width at half-maximum of the peak. Both the mean energy and errors are overlaid on the simulated spectra of Fig. 4b. These values are the signature used to derive the curvature of the evolving radial ion profile (see the following subsection). At bunch separations around 10 ns, multiple focal lines appear in a small energy range in the spectra, leading to a systematic increase in the average full-width at half-maximum. At the shortest timescales, a large fraction of the driving-probe-bunch charge is lost and hence the identification of the peaks in the spectra carries an associated higher uncertainty.

## Derivation of ion-channel-profile parameters

The first two experimental signatures—(1) the modification of the energy slice that is maximally focused by the post-plasma imaging optics due to the betatron mismatch, and (2) the oscillations of the r.m.s. transverse size of the trailing probe bunch—are a result of the motion of electrons in the probe bunches as they propagate in the plasma:

In the first experimental signature, an electron that propagates in the linear portion of the wakefield, that is, at the head of the driving probe bunch, undergoes transverse oscillations due to the focusing force provided by the wakefield. In the presence of a parabolic transverse-plasma-density profile, $n(r) = n_0(1 + \alpha r^2)$, the focusing force at a given longitudinal slice for a uniform density profile is modified by the factor $(1 + \alpha r^2)$. As such, the energies of the longitudinal slices that exit the plasma having acquired the appropriate betatron phase to correspond to bands of raised intensity observed on the spectrometer depend on $\alpha$ through the relation

$$\Delta_{\alpha,i} = \Delta_{0,i}(1 + \alpha r^2)^{2/3},$$

where $\Delta_{\alpha,i}$ and $\Delta_{0,i}$ are the deceleration of slice $i$ in a plasma with a non-zero and zero curvature, respectively. This enables reconstruction of the curvature as a function of the separation between the leading and probe bunches by fitting to the difference in energy between the band of raised intensity for the perturbed case and that in the equivalent unperturbed case.

In the second experimental signature, the trailing bunch has a low $O$(mm mrad) emittance and is focused to a centimetre-scale $\beta$-function at the entrance of the plasma; hence, it has a transverse size of approximately 8 μm, which is much smaller than even the steepest ion channel found in the experiment, where an increase of approximately 5% is predicted in the size of the trailing bunch and a doubling in density from the value on axis occurs at a radial position of approximately 40 μm. The trailing probe bunch therefore experiences limited changes to its off-axis focusing force due to the presence of the parabolic channel (confirmed in PIC simulations) and its divergence as a function of the on-axis plasma density follows the relation

$$\sigma_{x'} \propto |k_\beta \sin(k_\beta L_p)|$$

for a fixed plasma length $L_p$, where $k_\beta = \omega_\beta/c$. The plasma length is assumed to be constant over the $O$(100 ns) timescale considered here. The divergence of the bunch at the plasma exit directly relates to the r.m.s. transverse size measured in the plane of the electron spectrometer. Bunches with minimal and maximal divergence at the exit of the plasma can be focused to both small and large sizes and high and low intensities, respectively, at the scintillating screen. Therefore, the measured oscillations in the transverse size of the trailing probe bunch (Fig. 3a) can be correlated to extrema of the divergence of the trailing probe bunch at the plasma exit. This enables reconstruction of the on-axis plasma density as a function of the separation between the leading and probe bunches.

As the two experimental signatures are decoupled, the relevant equations can, therefore, be solved independently through numerical fitting of each equation to the pertinent experimental observable, that is, the energy of the betatron-mismatch bands for $\alpha$ and the r.m.s. transverse beam size for $n_0$ (Fig. 3a). The fitted values of the parabolic channel (and the associated fitting errors) are shown in Fig. 3b.

### 6D beam reconstruction

For accurate modelling of the plasma acceleration process, robust measurements of both the transverse and longitudinal phase spaces are required. A series of 11 quadrupoles downstream of the plasma cell was used to transport the electron bunches to an X-band transverse deflection structure (X-TDS)[42], where the beam was streaked onto a cerium-doped gadolinium aluminium gallium garnet screen for measurements of the longitudinal phase space. The energy-dispersed axis of the longitudinal phase space was provided by a dipole located between the X-TDS and the screen. The total length of the bunch was approximately 195 μm with a peak current of 1.5 kA for the leading bunch and 420 μm with a peak current of 1 kA for the unscraped probe bunch. The X-TDS has the feature of being able to streak in all transverse directions. As such, it was possible to derive slice information of the horizontal and vertical planes of the beam by streaking in the vertical and horizontal directions, respectively. Slice emittance measurements were performed in both planes for the leading and probe bunches, providing beam size and emittance information for every 8 μm slice. The X-TDS was only operated with non-plasma-interacted bunches and relaxed beam focusing due to the complexity of transporting high-divergence bunches the full distance (33 m) from the plasma to the X-TDS measurement screen.

### Particle-in-cell simulations

The 3D quasi-static PIC code HiPACE++ (ref. [43]) was used to simulate the full evolution of the beam–plasma interaction. The input beam was generated based on the 6D phase-space information of the experimentally characterized beams. It was modelled with $2 \times 10^6$ constant-weight macro-particles. A 32-mm-long flat-top plasma-density profile of height

$1.75 \times 10^{16}$ cm$^{-3}$ was estimated based on density measurements (see above). The plasma was sampled with 16 particles per cell. A simulation box of size $600 \times 600 \times 480$ μm$^3$ (in $x \times y \times \xi$, where $\xi = z - ct$ represents the co-moving frame) was resolved by a grid of $512 \times 512 \times 512$ cells, evolved with a constant time step of $4.5\omega_p^{-1}$, where $\omega_p$ is the plasma frequency.

## Data availability

The data presented in this paper and the other findings of this study are available from the corresponding author upon reasonable request.

## Code availability

All codes written for use in this study are available from the corresponding author upon reasonable request.

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

**Acknowledgements** We thank M. Dinter, S. Karstensen, S. Kottler, K. Ludwig, F. Marutzky, A. Rahali, V. Rybnikov, A. Schleiermacher, the FLASH management, and the DESY FH and M divisions for their scientific, engineering and technical support. We also thank C. Benedetti and C. B. Schroeder for their input on long-term-plasma-evolution theory. We thank W. Leemans for his help finalizing the manuscript. We also thank R. Jonas and K. Klose for developing the 1.3 GHz phase shifters to enable the diagnostic. This work was supported by Helmholtz ARD and the Helmholtz IuVF ZT-0009 programme, as well as the Maxwell computational resources at DESY. This work was supported in parts by a Leverhulme Trust Research Project grant no. RPG-2017-143. We acknowledge the use of the UCL Kathleen High Performance Computing Facilities (Kathleen@UCL), and associated support services, in the completion of this work.

**Author contributions** J.C., R.D. and J.O. conceived the experiment. J.C. wrote all data-taking scripts. J.B., R.D. and G.L. performed the experiment with help from J.C., M.J.G., P.G.C., C.A.L., S. Schröder, and S.W. J.C. performed all data analysis. Simulations were performed by J.C. with the help of S.D. and J.B. The manuscript was written by J.C. and R.D. with assistance from B.F., C.A.L., J.O., R.J.S. and M.W. The 6D beam reconstruction was performed by P.G.C. S.W. oversaw the technical development of the experimental infrastructure. M.J.G. and G.L. performed the plasma-density characterization. S.D. implemented parabolic-channel functionality to HiPACE++. R.J.S. conceived the illustration in Fig. 1 with help with its creation from S.D. G.B. provided crucial theoretical insights into long-term plasma evolution. C.A.L. quantified the timescale of the on-axis ion-density peak and wrote the equivalent methods section. S. Schreiber developed the 1.3 GHz bucket-jumping routine essential for the diagnostic. M.T. and J.C. demonstrated the density independence of the betatron-mismatch bands. R.D. and J.O. supervised the project and personnel. J.C. was supervised by M.W. All authors discussed the results in the paper.

**Funding** Open access funding provided by Deutsches Elektronen-Synchrotron (DESY).

**Competing interests** The authors declare no competing interests.

**Additional information**
**Correspondence and requests for materials** should be addressed to R. D'Arcy.

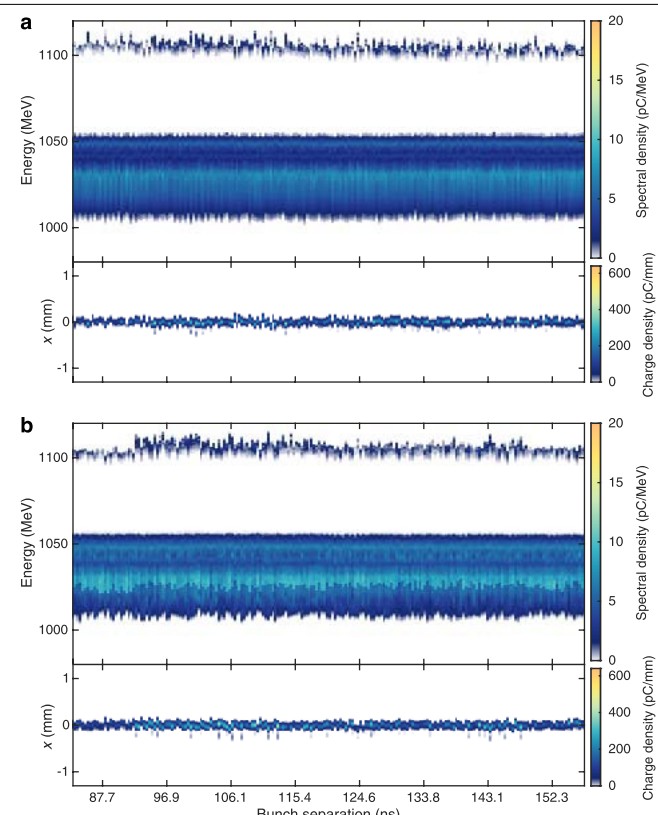

**Extended Data Fig. 1 | Continued recovery time scan (as in Fig. 2).**
**a**, The energy spectra and transverse distributions of the probe bunches after interaction with an unperturbed plasma. The results are those from a dataset taken shortly after that of Fig. 2, extending the data range from 87.7 ns to 152.3 ns in 9.23 ns steps. **b**, The same as in **a** but after interaction with a plasma perturbed by the leading bunch. Imperfections in the procedure used to subtract the overlapping spectra of the leading bunch from the driving bunch (see Methods) lead to small systematic differences between the energy spectra of **a** and **b** at low driver energies. Some trailing-bunch charge is lost at higher energies for the shortest bunch separation due to clipping in the quadrupoles. The similarity of the spectra and distributions plus the permanence of approximately zero residuals (see Fig. 2c) confirm that i) the ion motion has fully decayed and ii) no other perturbative plasma effects arising from the leading wakefield, which would cause deviation of the perturbed from the unperturbed case, occur within the measured timescale. Furthermore, according to PIC simulations such experimental uncertainties translate to a 1% difference in the longitudinally-integrated plasma density experienced by the probe bunch.

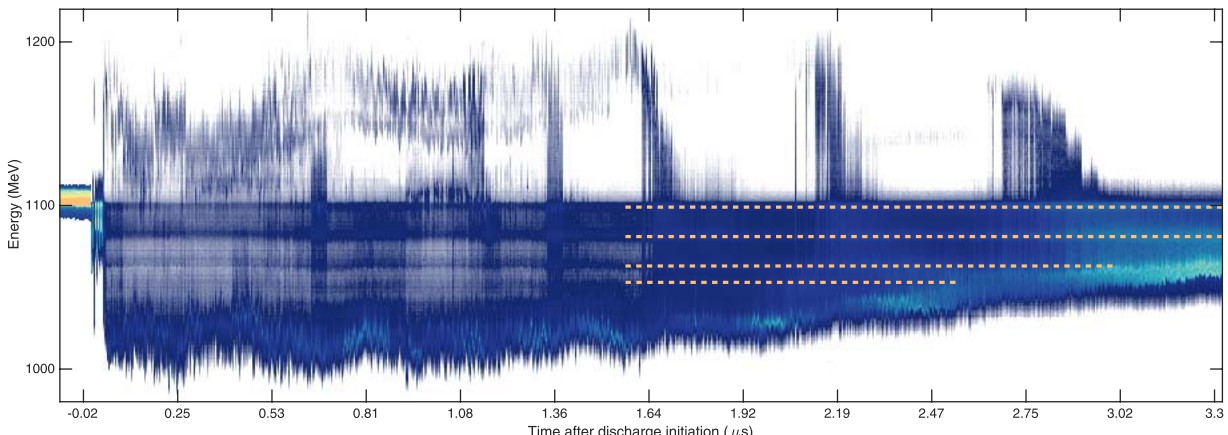

**Extended Data Fig. 2 | Constant betatron-mismatch bands for variable plasma density.** Energy spectra of the unscraped probe bunch of a working point similar to that of Fig. 2 as a function of delay after the start of the plasma-generating high-voltage discharge. The timing is scanned over a ~3 μs range, corresponding to a density decay of ~$10^{17}$ to ~$10^{15}$ cm$^{-3}$. The far-left spectra of high intensity at $t < 0$ represent discharge times after the beam has already traversed the capillary—effectively a *plasma off* state. The betatron-mismatch bands of increased intensity at constant energy— highlighted by orange dashed lines—indicate that the betatron oscillations of the driving-beam head are independent of plasma density and may therefore be used as a signal for the curvature of the channel.