## [Peer Review File · Nature]

Supplementary information

**Recovery time of a plasma-wakefield
accelerator**

In the format provided by the
authors and unedited

Peer Review File

Manuscript Title: Recovery time of a plasma-wakefield accelerator

Reviewer Comments & Author Rebuttals

Reviewer Reports on the Initial Version:

Referee #1 (Remarks to the Author):

The paper titled "Recovery time of a plasma wakefield accelerator" presents measurements and simulations of the plasma evolution for later times than previous work, which shows that MHz PWFA repetition rates are possible. High repetition rate is important for applications of the technology, such as high energy physics colliders that require high luminosity. This work extends the work of references 16 and 17, which show measurements at earlier times using optical techniques. The measurements are technically sound and really quite beautiful. The paper is extremely well written, with results presented clearly and concisely. I have no questions on the techniques or interpretation of the data. The one question I have is on the repetition rate. Although the temperature rise from a single drive bunch is negligible, at Mhz repetition rates the temperature will increase. How much? Will it affect the wake generation and acceleration? Will the structure even survive? Without discussion on getting rid of the heat during continuous operation, I'm not sure a repetition rate can be claimed.

The only other question to be answered is whether this topic should be published in Nature. From the Nature publishing website: "There should be a discernible reason why the work deserves the visibility of publication in a Nature Portfolio journal rather than the best of the specialist journals." There is inevitably subjectivity in this but allow me to elucidate my thought process on this. There is no doubt that operating at a high repetition rate is critical for PWFA. This paper is not the end of that discussion but is a big step in this area. It is of interest to anyone in the area of accelerator technology and certain applications such as high energy physics. Actually demonstrating MHz repetition rate or other of the pressing issues in PWFA seems more relevant to the visibility of Nature so my inclination would be that it belongs in the best of the specialist journals. However, the measurements presented are an important step in realizing the potential of PWFA machines, so one could argue publication in Nature is warranted.

Referee #2 (Remarks to the Author):

This manuscript describes the results of a plasma wakefield accelerator (PWFA) experiment carried out at FLASHForward to study the recovery time of a PWFA plasma source. This is important for trying to determine the theoretical upper limit to the repetition rate of PWFAs. The experiment was well conceived and executed. The data is abundant, of high quality, and is well explained by the proposed model of ion motion. The analysis of the data is sound, although the various assumptions that go into it are subtle and somewhat difficult to keep track of upon the first read through. The conclusions drawn about this particular experiment, operating in this particular parameter regime, are justified and of significant interest to the community. It is an interesting example within the realm of typical parameter space for modern basic research experiments. However, the authors go too far in extending the conclusions drawn from this one experiment to make overly general claims about the maximum repetition rate of all PWFA plasma sources and gloss over or ignore entirely other likely limiting factors, such as plasma heating. Even good beam loading can still leave 25-50% of the energy extracted from the drive beam in the plasma, and at high repetition rates this could lead to significant heating of the plasma source, which in turn could alter the density profile or cause other engineering problems. Further, even when considering only the average density profile of the plasma, this experiment does not consider extremely exact recovery of the original

density profile. It may be that it only takes $O(10\text{ ns})$ for recovery to within a few percent of the original average density after each shot, but if operating at the high repetition rates implied by this recovery time, i.e. $O(100\text{ kHz})$, one would quickly notice the average density drifting substantially. The fact that this study does not actually provide sufficient evidence to justify the overly bold and general claims regarding PWFA plasma source recovery time and maximum repetition rate is reason that it is not a good fit for Nature. Rather, it would be more appropriate for a publication like Nature Communications or Nature Physics. With that said, it is an excellent paper of a high caliber and sure to be of interest to the community. Below follow comments and suggestions for further improvement.

Main Manuscript:

- In the final paragraph on page 1, the authors distinguish this experiment from previous observations of ion motion following the passage of a PWFA beam by claiming theirs is “direct”, yet it seems no more or less direct than those other experiments that used an optical probe. This feels like an inaccurate claim, but moreover an unnecessary attempt at distinguishing the significance of this work from the others. The most significant distinction is just the timescale that was probed, which was sufficiently long so as to observe the average density recovery of the plasma.
- Figure 1: While the caption does claim this just to be an illustration, there should be more emphasis put on the fact that this is very “cartoony” in nature. For example, the central image does not show the beams or the wake to scale. Nor does it accurately show the damping rate of the wakes following the leading bunch. It gives the impression that the probe bunches are traveling in the nonlinear wake produced by the leading bunch, although that should die off well within the first nanosecond.
- On page 2 when the experiment is being described, it should be made clear that the expectation is that the leading bunch is driving a nonlinear wakefield (right?), as is the probe driver. In fact, it would be good to describe this in some detail. Is it a highly nonlinear blowout wake, for example?
- On page 2, paragraph 2 it should be made clear that it is the *average* leading bunch spectrum that gets subtracted from the spectrum of each individual shot.
- On page 2, paragraph 2 it could be made more clear that the gradual changes to the unperturbed plasma are simply in the nature of the capillary discharge plasma source, otherwise this may be mysterious and vague to the readers outside of or new to the field. The Methods section basically covers this, but it is a small change that could quickly dispel potential confusion from the current wording.
- Fig. 2: Though the energy change and transverse size of the bunches do seem to approach the unperturbed values by $\sim 63\text{ ns}$, the data does not continue much beyond that. It would be far more satisfying to see that there aren’t subsequent changes or oscillations at longer timescales, even if at smaller amplitude. Granted, the data likely doesn’t exist, but this comes back around to the problematically strong claims made in the conclusions from this study.
- On page 3, paragraph 1, a few other challenges for PWFA repetition rate are mentioned here, which is good, although heating should probably be in there, too, as it may be even more significant than the issues already listed. For that matter, could a significant rise in the initial temperature of the plasma (ions and/or electrons) even affect the ion evolution model?
- Fig. 3: Once again, the data seems to be a little short of the timescale that would be even more interesting and potentially convincing. Here, we don’t even reach the average density recovery time ($\sim 63\text{ ns}$, as claimed in Fig. 2), which is too bad.

- Fig. 4.a: This subfigure could be nicely enhanced by the addition of an overlaid lineout of the central axis density profile.
- Fig. 4.b: The agreement between data and simulation is the least good here, though it's not particularly bad. It might be nice to comment on why the experiment seems to show a pinch at ~ 18 ns but the simulation does not.
- Fig. 4.c: It might be good to remind the reader somewhere that the data and simulation don't agree below ~ 5 ns because this is where the data was less clean due to significant charge loss, ala the gray data points in Fig. 3.b. Thus, the disagreement can be mostly disregarded. (Right?)
- In the concluding paragraph, and throughout the paper, the term "recovery time" is used, but this is never precisely defined. As I stated above, what is actually meant is that the average density of the plasma near the beam axis returns to within X% of the original value.
- General note: It seems like it should be possible to estimate the betatron phase advance of the probe trailing beam for the unperturbed case, and then quantify the change in phase advance due to plasma perturbation. This would be very interesting to see, rather than just showing the transverse beam size. It feels like a more precise way to quantify this important observable.

Methods:

- On page 7, in the "Timescale for the formation..." section, it should be mentioned that the force equation applies only for the ions with an initial radius of less than the beam radius. Following onto that, would it be true that the spike effectively lasts longer than one would assume if only considering those particular ions due to the ions at larger radii reaching the central axis at later times? Since it seems that certain aspects of the experimental data were explained by interactions with a long-lasting ion spike, perhaps this is worth some elaboration.
- On page 7, in the "Origin of the density-independent..." section, references should be included to justify the focusing strength model that is used here.

Referee #3 (Remarks to the Author):

The manuscript investigates the fundamental repetition rate limit for plasma wakefield accelerators. The topic is of significance to the accelerator science community because it aims to address one of the challenges of plasma acceleration, i.e. the challenge of transitioning from a single-bunch experiment to a high-rep rate facility. The experimental approach is certainly valid and the data quality is high. However, I have two main objections to the proposed conclusions. Below I will provide my critique in detail.

1. Fundamental limit

The word "fundamental" is used in the abstract of the paper, implying that that the manuscript investigates the fundamental repetition rate limit. The authors never explain the "fundamental" nature of their findings.

Why is the recovery time of 63 ns viewed as "fundamental" by the authors? The authors propose that the plasma recovery time is determined by the speed, with which the plasma ion density spike is dissipated, i.e. the ion acoustic wave group velocity, $v \sim \sqrt{T_e/m_i}$. Already, one can argue that this velocity is not "fundamental", namely, it is not determined by fundamental physical constants. For example, one can explore a hotter plasma, i.e. increase T_e , or as authors propose, decrease the ion mass, m_i . There might be other ways. The authors write: "The timescales of the on-axis peak and refilling of the channel are also expected to depend on the temperature of the charge carriers, wakefield strength, and initial plasma-electron

density." In any case, the recovery time must also contain some length scale, L , as in time $t = L/v$. The authors never discuss what is the origin of this length scale L , which results in time $t = 63$ ns. Is it related to the plasma column geometry, in which case the rate limit is certainly not fundamental?

2. Repetition rate.

It is very hard for me to accept a rate limit estimate based on two-bunch measurements. A rate implies something that is steady state, but two bunches give only one time interval. Are there no cumulative effects at all? Let me examine the proposed rate limit, 15 MHz. From Figure 1a one can conclude that the drive bunch was decelerated by approximately 25 MeV in the plasma, corresponding to the deceleration rate of 0.5 GeV/m on average. The drive bunch charge, as reported in the manuscript, is about 0.5 nC. Thus, a single drive bunch leaves behind about 12 mJ of energy in the form of a plasma wake. Let us assume that the trailing bunch carries away 50% of this energy due to acceleration, which would make this plasma acceleration process 50% energy efficient. This efficiency assumption is very optimistic and much greater than what is reported in the paper. Thus, the remaining 50% of energy, ~ 6 mJ, has to be dissipated prior to the next bunch arrival. At 15 MHz rate, this corresponds to 90 kW of dissipated power. This is a tremendous challenge for a 1.5-mm-diameter, 50-mm-long capillary. The total internal wall area of the capillary is about 200 mm^2 . Thus, in a steady state operation at a 15 MHz repetition rate, the power density on the capillary wall would be $\sim 500 \text{ W/mm}^2$. In "warm" (non-superconductive) systems, the state-of-the-art heat absorption is limited at $\sim 10 \text{ W/mm}^2$. The authors do not discuss the power dissipation challenge at all. The only statement that I have found in the manuscript is: "In order to realise this rate at future facilities, the technical challenge of creating a plasma source capable of both mitigating the slow expulsion of plasma into vacuum and counteracting recombination must be met." It sounds like the authors are proposing to dissipate the 90 kW power by expelling plasma out of capillary ends. The future plasma channels of interest may have to be 1-m long (as opposed to 50-mm in the manuscript) in order to provide GeV's of acceleration. The power dissipation should also increase in proportion. The power absorption and dissipation rates in capillaries may give the actual fundamental limit on repetition rates.

Overall, I view this manuscript as a good initial experimental work but far from being decisive in determining the fundamental repetition rate limits.

The manuscript could be improved by discussing the power balance and by discussing any plasma and capillary cumulative effects from passing bunches. Numerically, it would be good to see how the plasma wake is thermalized, and how much of this wake energy is transferred to the surrounding environment.

Author Rebuttals to Initial Comments:

First and foremost, we would like to thank all the referees for their review of our paper and their constructive comments on how to enhance it. This has led to several improvements throughout the manuscript (see detailed direct replies below).

There are two predominant queries from all three referees, namely the semantic interpretation of a “fundamental (repetition rate) limit” and the likely future challenges arising as a consequence of scaling up to high repetition rates. With hindsight, we realise that we were not careful enough with the original wording of our paper on both counts and, therefore, an understandable confusion has arisen. We believe that clarification of the first point is inextricably linked to that of the second.

In the context of this result, when we used the term *fundamental limit* we were referring to the upper limit placed on repetition rate by the need to allow the motion of the perturbed plasma to reach its natural end such that, to a subsequent driver, it is indistinguishable from the unperturbed plasma. This upper limit is considered to be fundamental in that it is defined by a basic physics process that cannot be prematurely stopped and is, therefore, isolated from all subsequent engineering problems. Through our experimentation, we have characterised for the first time this fundamental physics process and the upper limit it places on inter-bunch repetition rate. We have, therefore, rephrased the paper to refer to an *upper limit* determined by the *fundamental physics process* of plasma motion over long timescales.

Due to the experimental results of this study, long-term plasma evolution can be considered as the most basic repetition-rate constraint on structures in future facilities. As modern conventional accelerators operate in either continuous-wave (CW) or burst mode, it is ambiguous to mention a repetition rate without also specifying the bunch-train pattern of the accelerator. By framing (and titling) our work in the context of a recovery time rather than a repetition rate, we naively overlooked this ambiguity when converting to a repetition rate. Consequently, we have redefined the repetition rate as *inter-bunch* throughout the paper in the hope that this makes clear that we do not automatically equate this upper limit to the total number of bunches accelerated per second.

Now that our paper establishes the plasma recovery time, the natural next step for the field is to consider how to approach this upper limit in a real accelerator. At this point—and as rightly pointed out by the referees—it is necessary to consider the question of challenges arising from, for example, the accumulation of heat within the plasma from the

energy dissipated in each acceleration event. Particle-in-cell simulations have shown that plasma temperatures up to $O(100 \text{ eV})$ are unlikely to significantly modify the plasma-wakefield properties [see e.g. Lotov, *Phys. Rev. ST Accel. Beams* **6**, 061301 (2003), Jain *et al.*, *Phys. Plasmas* **22**, 023103 (2015), Esarey *et al.*, *Phys. Plasmas* **14**, 056707 (2007)]. Temperature ranges higher than this are yet to be simulated. Furthermore, on a macroscopic level, cumulative heating of the plasma over many bunches could lead to damage of the plasma source if left unchecked.

One approach to mitigating any deleterious effects of cumulating heating would be to consider the optimum temporal structure of accelerated bunches. For example, limiting changes to the wakefield properties may be the crucial factor in determining the number of bunches in a single bunch train, while the number of bunch trains accelerated per second may be limited by damage to the plasma source. With this in mind, an $O(10 \text{ MHz})$ bunch frequency within bunch trains separated by a large temporal gap, similar to that proposed for both the ILC and CLIC colliders, could best serve the operation of plasma-based linear colliders as well as free-electron lasers. A further benefit of redefining our result as an inter-bunch separation is that it intentionally leaves open for consideration the question of how bunch-train patterns in future plasma-based facilities may look. In parallel to this, increasing the durability of the plasma-accelerator infrastructure may be possible through a paradigm shift in plasma-source technology to one designed specifically with high repetition rate in mind, for example HOFI channels [see e.g. Shaloo *et al.*, *Phys. Rev. E* **97**, 053203 (2018), Picksley, *et al.*, *Phys. Rev. E* **102**, 053201 (2020)]. For such sources the cooling dynamics would be entirely different but the wakefield-induced motion of the ions, as characterised in this work, would be unchanged.

These questions only now merit detailed investigation as a consequence of our result, for had the basic physics of plasma motion been shown to be such as to extend the recovery time of the plasma to, for example, many hundreds of microseconds, there would be no point in considering the possibility of engineering solutions for the implementation problems. We agree with the referees that addressing these open questions is crucial if a working accelerator based on these principles is to be constructed. As such, we have added an additional paragraph to the manuscript which addresses the need to further consider bunch-train patterns and develop plasma-source technology, in particular to deal with cumulative heating. We hope that, with these additions and clarifications, the referees will agree with us that this an important and novel result, worthy of publication in *Nature*.

Quoted below are the referees' comments (in **blue**) and our response to each of them (in black). Changes to the manuscript are indicated in **bold** in this response letter. In the accompanying edited manuscript [titled '*Manuscript (changes tracked)*'], any added text is highlighted as **bold, underlined, blue type** with any text removed indicated with a ~~strikethrough~~.

Referee #1:

The paper titled "Recovery time of a plasma wakefield accelerator" presents measurements and simulations of the plasma evolution for later times than previous work, which shows that MHz PWFA repetition rates are possible. High repetition rate is important for applications of the technology, such as high energy physics colliders that require high luminosity. This work extends the work of references 16 and 17, which show measurements at earlier times using

optical techniques. The measurements are technically sound and really quite beautiful. The paper is extremely well written, with results presented clearly and concisely. I have no questions on the techniques or interpretation of the data.

We thank the referee for this positive summary of our work as well as the recognition of the important role it will play in the wider community.

The one question I have is on the repetition rate. Although the temperature rise from a single drive bunch is negligible, at Mhz repetition rates the temperature will increase. How much? Will it affect the wake generation and acceleration? Will the structure even survive? Without discussion on getting rid of the heat during continuous operation, I'm not sure a repetition rate can be claimed.

As discussed above in the introductory paragraphs, we agree with the referee on the subject of defining a repetition rate and acknowledge that this ambiguity is due to a lack of discussion on the necessary next steps within this new field, expanded by the results of this submission. As such, we have reworded the manuscript appropriately—**specifically by adding a discussion on bunch-train patterns, heating effects, engineering questions, and next steps at the end of the manuscript.**

With regard to how an increase in plasma temperature may affect wake generation and acceleration, PIC simulations on this topic have been performed in other works [see e.g. Lotov, Phys. Rev. ST Accel. Beams **6**, 061301 (2003), Jain *et al.*, Phys. Plasmas **22**, 023103 (2015), Esarey *et al.*, Phys. Plasmas **14**, 056707 (2007)]. The results of these studies suggest that plasma temperatures of order 100 eV reduce the electron density spike, and hence wakefield amplitude, at the very rear of the wakefield cavity. These are not expected to significantly modify the acceleration process of the accelerating bunch, however, as the wakefield over the vast majority of the cavity (more importantly where the accelerating bunch will be placed) will be negligibly affected. Beyond this temperature range—currently unexplored in PIC simulations—the wakefield structure may be modified. If such is the case, the number of bunches in a bunch train, with inter-bunch spacing equal to the recovery time, would be selected in order to mitigate these effects. **We have included a summary of this information in the new discussion section at the end of the manuscript.**

The only other question to be answered is whether this topic should be published in Nature. From the Nature publishing website: “There should be a discernible reason why the work deserves the visibility of publication in a Nature Portfolio journal rather than the best of the specialist journals.” There is inevitably subjectivity in this but allow me to elucidate my thought process on this. There is no doubt that operating at a high repetition rate is critical for PWFA. This paper is not the end of that discussion but is a big step in this area. It is of interest to anyone in the area of accelerator technology and certain applications such as high energy physics. Actually demonstrating MHz repetition rate or other of the pressing issues in PWFA seems more relevant to the visibility of Nature so my inclination would be that it belongs in the best of the specialist journals. However, the measurements presented are an important step in realizing the potential of PWFA machines, so one could argue publication in Nature is warranted.

We thank the referee for their openness in explaining their opinion. We submit that the possibility of an actual demonstration of sustained MHz plasma acceleration with acceptable beam parameters for application can only be achieved with a well-engineered plasma-source solution featuring appropriate cooling, at an accelerator facility with the possibility to explore the required bunch spacing and repetition rate—something that lies many years, possibly even decades, and many millions of euros/dollars/pounds, in the future. Our contention is that our results demonstrate for the first time that a meaningful

effort in the direction of such a demonstration is now worthwhile. As such, they open up a new regime of plasma acceleration with impact well beyond the immediate field, into photon science, engineering, and the multitude of applications that rely on high-energy-particle and x-ray beams.

Referee #2:

This manuscript describes the results of a plasma wakefield accelerator (PWFA) experiment carried out at FLASHForward to study the recovery time of a PWFA plasma source. This is important for trying to determine the theoretical upper limit to the repetition rate of PWFAs. The experiment was well conceived and executed. The data is abundant, of high quality, and is well explained by the proposed model of ion motion. The analysis of the data is sound, although the various assumptions that go into it are subtle and somewhat difficult to keep track of upon the first read through. The conclusions drawn about this particular experiment, operating in this particular parameter regime, are justified and of significant interest to the community. It is an interesting example within the realm of typical parameter space for modern basic research experiments.

We thank the referee for this positive evaluation of our paper.

However, the authors go too far in extending the conclusions drawn from this one experiment to make overly general claims about the maximum repetition rate of all PWFA plasma sources and gloss over or ignore entirely other likely limiting factors, such as plasma heating.

To draw such conclusions was not our intention but our wording was certainly unclear on this point. We believe that the ambiguity lay with our use of the term “repetition rate” without any qualifier on bunch-train structure, which could have led to the assumption that we were implying 15 MHz CW operation—for which plasma heating is indeed likely to be an obstacle. As such, **we have rephrased the limits and rates of our work throughout in an attempt to clarify this and have added a discussion on heating to the end of the manuscript.** In short, we completely agree that there are many other factors that must be taken into consideration before a MHz plasma-based accelerator can be constructed and submit that it is the results of this paper that now motivate the detailed investigation of such questions.

Even good beam loading can still leave 25-50% of the energy extracted from the drive beam in the plasma, and at high repetition rates this could lead to significant heating of the plasma source, which in turn could alter the density profile or cause other engineering problems. Further, even when considering only the average density profile of the plasma, this experiment does not consider extremely exact recovery of the original density profile. It may be that it only takes $O(10\text{ ns})$ for recovery to within a few percent of the original average density after each shot, but if operating at the high repetition rates implied by this recovery time, i.e. $O(100\text{ kHz})$, one would quickly notice the average density drifting substantially. The fact that this study does not actually provide sufficient evidence to justify the overly bold and general claims regarding PWFA plasma source recovery time and maximum repetition rate is reason that it is not a good fit for Nature. Rather, it would be more appropriate for a publication like Nature Communications or Nature Physics. With that said, it is an excellent paper of a high caliber and sure to be of interest to the community.

We agree with the referee that this is an important consideration. As stated in the summary section at the beginning of this response letter, the absence of a discussion on cumulative heating from the first version of the manuscript was an oversight on our part. We agree with the referee that the accumulation of heat from operation at sustained and high rates of repetition may cause problems if left unchecked. For example, the bunch-train length may be limited by temperature effects on the wake structure and the number of bunch trains per second may be limited by engineering challenges arising from

accumulate heating. It may be possible to manage both effects through careful optimisation of the bunch-train pattern, as proposed at both ILC and CLIC. For the latter effect, cumulative heating of the plasma source may be mitigated through, for example, a paradigm shift in plasma source technology e.g. Shaloo *et al.* Asking and answering these questions is the next logical step for the field. In the updated version of the manuscript we mention next steps for the field as a result of possible challenges in the **discussion paragraph added towards the end of the manuscript.**

Furthermore, and as stated in our response to Referee #1, we submit that the possibility of actually demonstrating sustained MHz acceleration with application-worthy engineering solutions is many years away. Our contention is that our result, which implies that such a long-term effort is worthwhile, is of great and topical interest to the accelerator, particle physics, and photon science community, as the referee notes. We believe this result will catalyse much future work and is therefore worthy of publication in *Nature*.

Below follow comments and suggestions for further improvement.

Main Manuscript:

In the final paragraph on page 1, the authors distinguish this experiment from previous observations of ion motion following the passage of a PWFA beam by claiming theirs is “direct”, yet it seems no more or less direct than those other experiments that used an optical probe. This feels like an inaccurate claim, but moreover an unnecessary attempt at distinguishing the significance of this work from the others. The most significant distinction is just the timescale that was probed, which was sufficiently long so as to observe the average density recovery of the plasma.

We respectfully disagree with the referee that the results presented here are simply an extension of the optically probed timescales of previous experimentation. The novel diagnostic technique developed for this experiment can be considered more direct than an optical probe measuring a refractive index perturbation in two senses. The first is that the probe bunch directly samples the perturbed ion-density profile in the form of the generated electric fields, whereas an optical probe can only infer the ion-density profile from the increased density of electrons attracted to the associated ion-density spikes and the thereby altered refractive index. The second is that the probe-bunch methodology makes a direct measurement of the deleterious effects of ion motion on wakefield excitation by employing the typical two-bunch acceleration scheme as a diagnostic in itself. In doing so, any effect that can negatively impact the upper limit of the repetition rate of this process is automatically explored.

We do concede, however, that the use of the word “direct” may distract the reader. As such, **we have removed any mention of ‘directness’ from the opening page.**

Figure 1: While the caption does claim this just to be an illustration, there should be more emphasis put on the fact that this is very “cartoony” in nature. For example, the central image does not show the beams or the wake to scale. Nor does it accurately show the damping rate of the wakes following the leading bunch. It gives the impression that the probe bunches are traveling in the nonlinear wake produced by the leading bunch, although that should die off well within the first nanosecond.

We thank the referee for this observation. In our original version of the figure, our aim was to demonstrate the concept of the probe process, namely that a later-arriving probe bunch would sample the plasma as perturbed by the leading bunch. The referee is of course correct, however, that the leading-bunch plasma-electron wake will have decayed

long before the arrival of the probe bunch. In order to not unintentionally mislead the general *Nature* reader, we have modified the figure to better ‘illustrate’ the evolution of the pertinent physics processes whilst maintaining the concept of the diagnostic method. We hope that **this modification to the figure, as well as relabelling as a ‘Conceptual representation’**, will meet the referee’s point of substance while still allowing us to give the reader the appropriate picture in their minds, which is important for the subsequent exposition.

On page 2 when the experiment is being described, it should be made clear that the expectation is that the leading bunch is driving a nonlinear wakefield (right?), as is the probe driver. In fact, it would be good to describe this in some detail. Is it a highly nonlinear blowout wake, for example?

The peak density of both the leading bunch and driving probe bunch are ~ 3 times that of the baseline (unperturbed) plasma density at the entrance to the plasma cell, with their local density expected to increase during evolution in plasma due to transverse mismatching. As the condition for a blown-out wake (bunch density $>$ plasma density) is met, the trailing probe bunch is expected to always propagate in a blown-out wake. Furthermore, an ion column—completely void of plasma electrons—has also been confirmed behind the driver in our short-timescale PIC simulations, the results of which were used to generate Fig. 4. To reflect this, **the qualifier *nonlinear* has been added to the text where the wakefield (driven by both the leading and probe bunches) is first introduced.**

On page 2, paragraph 2 it should be made clear that it is the *average* leading bunch spectrum that gets subtracted from the spectrum of each individual shot.

Once again we agree with the referee. This is made clear in the ‘Methods’ section but **has also now been clarified in the main body of the text** [Manuscript (tracked changes) — page 2, lefthand column, final paragraph].

On page 2, paragraph 2 it could be made more clear that the gradual changes to the unperturbed plasma are simply in the nature of the capillary discharge plasma source, otherwise this may be mysterious and vague to the readers outside of or new to the field. The Methods section basically covers this, but it is a small change that could quickly dispel potential confusion from the current wording.

We agree with the referee that the absence of this information in the main body of the text may lead to confusion to even an expert reader upon first pass. **This additional information has therefore been added** [Manuscript (tracked changes) — page 3, lefthand column, first paragraph].

Fig. 2: Though the energy change and transverse size of the bunches do seem to approach the unperturbed values by ~63 ns, the data does not continue much beyond that. It would be far more satisfying to see that there aren't subsequent changes or oscillations at longer timescales, even if at smaller amplitude. Granted, the data likely doesn't exist, but this comes back around to the problematically strong claims made in the conclusions from this study.

Indeed, the data requested by the referee does exist. It is taken at separations beyond that of Fig. 2 and is displayed in Extended Data Fig. 1 i.e. from 80–160 ns in the original submission. As can be seen from this data, the demonstration of consistent evolution of both the perturbed and unperturbed plasma over a subsequent timescale in excess of that of the original perturbation is in full support of the return of the perturbed plasma to an unperturbed state, thus supporting our conclusions. Due to the importance of this extended data range, **the 'Residuals' information previously contained within Extended Data Fig. 1 has been incorporated into Fig. 2 (see below), with the final part of Extended Data Fig. 1 removed.**

On page 3, paragraph 1, a few other challenges for PWFA repetition rate are mentioned here, which is good, although heating should probably be in there, too, as it may be even more significant than the issues already listed. For that matter, could a significant rise in the initial temperature of the plasma (ions and/or electrons) even affect the ion evolution model?

We agree with the referee on the topic of cumulative heating and acknowledge that its inclusion is required in order to complete the grander picture of the implication of our results. As such, **a discussion on engineering challenges has been moved towards the end of the manuscript and expanded from its original short format.**

With regards to the impact of changes to the initial plasma temperature on the ion-evolution model, we are glad that questions such as these may now be raised. We discuss on page 3 of the manuscript that the timescale of the outwardly and inwardly propagating ion acoustic waves are expected to depend on the initial plasma-electron temperature. Unfortunately we are unable to answer this question experimentally with the

data sets contained within this submission. However, and as mentioned in the original outlook paragraph towards the end of the manuscript (beginning “The success of the evolving-ion-channel model...”), experimental evidence for the dependency of the plasma motion on key parameters e.g. temperature, ion mass, etc. will form the next steps in this research.

Fig. 3: Once again, the data seems to be a little short of the timescale that would be even more interesting and potentially convincing. Here, we don't even reach the average density recovery time (~63 ns, as claimed in Fig. 2), which is too bad.

We entirely agree with the referee that extra data on longer timescales for Fig. 3 would be valuable. Although extended timescale data does not exist for Fig. 3, we would draw the referee's attention instead to Extended Data Figure 1 (part of which has now been brought into Fig. 2 in the main manuscript). At this second working point—with probe-bunch properties optimised to amplify the signatures of ion motion—the properties of the leading bunch entering the plasma were identical to those of the first working point (Fig. 2) to within diagnostic uncertainties. As such, we consider it highly improbable that the conditions under which Fig. 3 were produced would not be well represented by the longer times shown in Extended Data Fig. 1. Furthermore, the shorter timescales shown in Fig. 3 were selected to allow the behaviour of interest in the first 50 ns to be seen more clearly.

Fig. 4.a: This subfigure could be nicely enhanced by the addition of an overlaid lineout of the central axis density profile.

We thank the referee for this suggestion. Unfortunately, and after many iterations, we were unable to find a solution to incorporate the suggestion without either adding distracting complexity to the subfigure (e.g. through additional axes) or reducing the quality of the subsequent subfigures (e.g. through a reduction in scale to accommodate the additional information in a). Given the relation to the experimental information already contained in Fig. 3b, however, we hope that the narrative progression between the results of Figs. 3 and 4 is not diminished as a result of the exclusion.

Fig. 4.b: The agreement between data and simulation is the least good here, though it's not particularly bad. It might be nice to comment on why the experiment seems to show a pinch at ~18 ns but the simulation does not.

The source of the discrepancy between experiment and simulation at ~18 ns is due to charge loss from the highly divergent trailing probe bunch clipping in the capturing quadrupoles downstream of the plasma exit. This brief explanation is included in the caption of Fig. 3. However, we agree that it bears repeating, and **we have added a comment to this effect to the caption of Fig. 4.**

Fig. 4.c: It might be good to remind the reader somewhere that the data and simulation don't agree below ~5 ns because this is where the data was less clean due to significant charge loss, ala the gray data points in Fig. 3.b. Thus, the disagreement can be mostly disregarded. (Right?)

The referee's interpretation of the charge loss at short timescales is correct. **A comment to this effect has been added to the text** [Manuscript (tracked changes) — page 4, righthand column, penultimate paragraph] in order to prevent the general *Nature* reader from having to make similar assumptions about the source and impact of this

disagreement. Furthermore, **Fig. 4c has been reformatted to highlight the additional uncertainty associated with the results at the shortest separation—** also in grey for consistency with Fig. 3, as suggested by the referee.

In the concluding paragraph, and throughout the paper, the term “recovery time” is used, but this is never precisely defined. As I stated above, what is actually meant is that the average density of the plasma near the beam axis returns to within X% of the original value.

The recovery time was qualitatively defined at the bottom of the first paragraph on page 2. In order to place a value on the discrepancy of the average density of the plasma after the perturbation, the experimental uncertainties of Extended Data Figure 1 were incorporated into further particle-in-cell simulations of the experiment. The results of these simulations indicated that the longitudinally integrated plasma density was identical to that of the unperturbed case to within 1%. **This information has been added to the caption of Extended Data Figure 1** and we thank the referee for suggesting this improvement.

General note: It seems like it should be possible to estimate the betatron phase advance of the probe trailing beam for the unperturbed case, and then quantify the change in phase advance due to plasma perturbation. This would be very interesting to see, rather than just showing the transverse beam size. It feels like a more precise way to quantify this important observable.

We agree with the referee that this would be possible. However, we deemed there to be no advantage in using a derived quantity such as the betatron phase advance when we could characterise the physics adequately with a directly measured one i.e. the beam size, a quantity which is likely more accessible to a general readership, as is expected with *Nature*.

Methods:

On page 7, in the “Timescale for the formation...” section, it should be mentioned that the force equation applies only for the ions with an initial radius of less than the beam radius. Following onto that, would it be true that the spike effectively lasts longer than one would assume if only considering those particular ions due to the ions at larger radii reaching the central axis at later times? Since it seems that certain aspects of the experimental data were explained by interactions with a long-lasting ion spike, perhaps this is worth some elaboration.

We thank the referee for this observation. We agree that it is fair to assume some ions will indeed reach the axis later than our model approximates, thus somewhat extending the timescale of the on-axis ion spike. However, the approximation is already intentionally simplified in order to make an illustrative calculation based on experimental measurables i.e. the leading-bunch properties. As a result of this, we explicitly state in the manuscript that, without the inclusion of the contribution from the wake—the properties of which we cannot measure—this simplified calculation estimates an upper bound on the timescale. It is important to clarify this where the calculation is outlined in full, however, so **we have added the necessary text—specifically the approximate nature of the value—to the appropriate Methods section** [Manuscript (tracked changes)— page 7, lefthand [righthand] column, fifth [first] paragraph].

On page 7, in the “Origin of the density-independent...” section, references should be included to justify the focusing strength model that is used here.

The focusing-field strength quoted in the relevant Methods subsection is derived from linear wakefield theory. Although relatively direct, it is unfortunately not possible to include the derivation in full due to space constraints. However, as requested **references to the relevant fundamental theory used for the derivation have been added for clarity (see below), with the text expanded to include a more robust qualitative explanation of the derivations** [Manuscript (tracked changes) — page 7, righthand column, 3rd and 4th paragraph].

- Gorbunov, L.M., & Kirsanov, V.I., Excitation of plasma waves by an electromagnetic wave packet. Sov. Phys. JETP66.2, 290-294 (1987).
- Sprangle, P., et al., Laser wakefield acceleration and relativistic optical guiding. AIP Conference Proceedings. Vol. 175. No. 1. American Institute of Physics (1988).

Referee #3:

The manuscript investigates the fundamental repetition rate limit for plasma wakefield accelerators. The topic is of significance to the accelerator science community because it aims to address one of the challenges of plasma acceleration, i.e. the challenge of transitioning from a single-bunch experiment to a high-rep rate facility. The experimental approach is certainly valid and the data quality is high.

We thank the referee for this positive evaluation.

However, I have two main objections to the proposed conclusions. Below I will provide my critique in detail.

1. Fundamental limit

The word "fundamental" is used in the abstract of the paper, implying that the manuscript investigates the fundamental repetition rate limit. The authors never explain the "fundamental" nature of their findings. Why is the recovery time of 63 ns viewed as "fundamental" by the authors? The authors propose that the plasma recovery time is determined by the speed, with which the plasma ion density spike is dissipated, i.e. the ion acoustic wave group velocity, $v \sim \sqrt{T_e/m_i}$. Already, one can argue that this velocity is not "fundamental", namely, it is not determined by fundamental physical constants. For example, one can explore a hotter plasma, i.e. increase T_e , or as authors propose, decrease the ion mass, m_i . There might be other ways. The authors write: "The timescales of the on-axis peak and refilling of the channel are also expected to depend on the temperature of the charge carriers, wakefield strength, and initial plasma-electron density." In any case, the recovery time must also contain some length scale, L , as in time $t = L/v$. The authors never discuss what is the origin of this length scale L , which results in time $t = 63$ ns. Is it related to the plasma column geometry, in which case the rate limit is certainly not fundamental?

We are grateful to the referee for pointing out this ambiguity, which we believe arose due to our original wording. By use of the word "fundamental", we intended to imply an upper limit placed on the inter-bunch repetition rate by fundamental physics processes occurring after the driving of a wake i.e. long-term plasma evolution. The processes themselves cannot be ended through engineering improvements and, as such, define the maximum possible repetition rate of a plasma-wakefield accelerator. We have, therefore, **rephrased the paper to refer to an upper limit determined by the fundamental physics process of plasma motion over long timescales.**

This clarification helps to address the referee's next point regarding our quantification of the recovery time. It was not our intention to imply any particular significance to the 63 ns as this was simply the result of our particular experimental conditions. The inclusion of the paragraph towards the end of the original manuscript on scalings and dependencies (beginning "The success of the evolving-ion-channel model...") was meant to both

highlight this point as well as to lay out a roadmap for optimising this value for future facilities. On the naive assumption that this return to the unperturbed conditions is purely determined by the ion acoustic velocity, then it is of course simple to calculate how the 63 ns will change as a function of gas species, temperature etc. However, during the recovery period the plasma evolution is affected by many different processes—the development of an on-axis density peak and outwardly/inwardly moving ion acoustic waves—which are expected to have varying dependencies on the same experimental parameters. As PIC simulations are currently incapable of mapping this phase space for us, it is necessary to explore this experimentally. We intend to investigate this in future experiments.

Finally, on the question of pertinent length scales, the referee is correct to point this out. The recovery of the plasma is governed by the time taken for the ions to return to their original average position after the perturbation by the beam. This timescale is determined by the damping of the outwardly and inwardly propagating ion acoustic waves, which have a velocity, and thus this damping occurs over a characteristic length scale. This length scale is therefore not expected to depend on the plasma column/plasma source geometry.

2. Repetition rate.

It is very hard for me to accept a rate limit estimate based on two-bunch measurements. A rate implies something that is steady state, but two bunches give only one time interval. Are there no cumulative effects at all? Let me examine the proposed rate limit, 15 MHz. From Figure 1a one can conclude that the drive bunch was decelerated by approximately 25 MeV in the plasma, corresponding to the deceleration rate of 0.5 GeV/m on average. The drive bunch charge, as reported in the manuscript, is about 0.5 nC. Thus, a single drive bunch leaves behind about 12 mJ of energy in the form of a plasma wake. Let us assume that the trailing bunch carries away 50% of this energy due to acceleration, which would make this plasma acceleration process 50% energy efficient. This efficiency assumption is very optimistic and much greater than what is reported in the paper. Thus, the remaining 50% of energy, ~6 mJ, has to be dissipated prior to the next bunch arrival. At 15 MHz rate, this corresponds to 90 kW of dissipated power. This is a tremendous challenge for a 1.5-mm-diameter, 50-mm-long capillary. The total internal wall area of the capillary is about 200 mm^2 . Thus, in a steady state operation at a 15 MHz repetition rate, the power density on the capillary wall would be $\sim 500 \text{ W/mm}^2$. In "warm" (non-superconductive) systems, the state-of-the-art heat absorption is limited at $\sim 10 \text{ W/mm}^2$. The authors do not discuss the power dissipation challenge at all. The only statement that I have found in the manuscript is: "In order to realise this rate at future facilities, the technical challenge of creating a plasma source capable of both mitigating the slow expulsion of plasma into vacuum and counteracting recombination must be met." It sounds like the authors are proposing to dissipate the 90 kW power by expelling plasma out of capillary ends. The future plasma channels of interest may have to be 1-m long (as opposed to 50-mm in the manuscript) in order to provide GeV's of acceleration. The power dissipation should also increase in proportion. The power absorption and dissipation rates in capillaries may give the actual fundamental limit on repetition rates.

We thank the referee for broaching the important topic of repetition rate. As stated in our introduction at the beginning of this letter, some ambiguity has arisen due to our focus on a recovery time and a subsequent lack of care with the definition of rates within the paper. It is ambiguous to quote a repetition rate without also specifying the bunch-train structure of the accelerator. As the recovery time defines a minimum time between two consecutive acceleration events and therefore an upper limit on the *maximum* CW repetition rate, we have **redefined the repetition rate as *inter-bunch* throughout the paper to better represent this repetition-rate upper limit**. We hope that this makes clear that we do not automatically translate this inter-bunch spacing to the total number of bunches accelerated per second.

This clarification leads directly on to the second query of the referee. Here, we respectfully disagree that a rate implies something that is steady state. Indeed, modern conventional accelerators operate in either CW or burst mode. In the case of the former, the inter-bunch rate is equal to the number of bunches per second whereas for an accelerator operating in burst mode, the number of bunches accelerated per second is far smaller. Take for example the bunch structure proposed for CLIC (operating with a centre-of-mass energy of 3 TeV) [see e.g. Aicheler *et al.* <https://doi.org/10.2172/1120127>]. CLIC is expected to operate with a base radio-frequency (RF) of 12 GHz, as defined by the X-band accelerating structures and power supplies. However, due to the lifetime of the transverse wakefields stimulated by a single acceleration event, the inter-bunch separation is limited to 0.5 ns i.e. a repetition-rate upper limit of 2 GHz. From this primary building block—which is in principle analogous to our result but defined by completely different physics processes—the number of bunches per bunch train (312) and the separation between bunch trains (~120 ns) can be defined. In each case, these values are defined by engineering limitations; specifically breakdown rates of the metallic cavities and the cumulative heating of the cavities from the RF, respectively.

The heating calculation the referee makes is more than reasonable to arrive at an order-of-magnitude value but assumes CW operation at a rate defined by the recovery time. To suggest that future plasma-based facilities should be operated in such a way was not our intention. Indeed it is possible that the engineering challenges rightly pointed out by the referee may ultimately preclude operation at CW MHz. It may be possible to mitigate the cooling requirements through a shift in plasma-source technology away from that used in this experiment and towards that specifically designed with high repetition rates in mind e.g. HOFI channels [see e.g. Shaloo *et al.*, *Phys. Rev. E* **97**, 053203 (2018), Picksley, *et al.*, *Phys. Rev. E* **102**, 053201 (2020)]. However, even with novel engineering schemes, a limit may be placed on the bunch train length and number of bunch trains per second through heating. To address these challenges is the obvious next step for this field.

It is our belief that our result, which for the first time quantifies the plasma recovery time and elucidates the physics defining it, isolates the most basic aspect of the acceleration process—the wakefield generation and subsequent wakefield dissipation—from all the engineering problems associated with constructing a practical accelerator. As such, future engineering challenges fall outside the scope of this work. Our contention is rather that the result reported herein establishes that further work in the direction indicated by the referee is indeed worth pursuing in earnest. In order to acknowledge the important issues brought up by the referee, **we have added a paragraph to the end of the paper addressing these topics.**

Overall, I view this manuscript as a good initial experimental work but far from being decisive in determining the fundamental repetition rate limits. The manuscript could be improved by discussing the power balance and by discussing any plasma and capillary cumulative effects from passing bunches. Numerically, it would be good to see how the plasma wake is thermalized, and how much of this wake energy is transferred to the surrounding environment.

While we disagree with the referee's first sentence, which relates to different interpretations of the word "fundamental" (discussed in previous responses), we believe that we have partially addressed these concerns by our general statement at the beginning of this response letter **and the addition of a section on the importance of**

heat dissipation at the end of the manuscript. Although we agree that further expansion of the paper along the lines that the referee suggests would be appropriate, the constraints on the length of the paper, but more importantly our wish to concentrate specifically on the result we have obtained, preclude this. We believe that the paper as amended removes the sources of confusion highlighted by the referee and focusses now on our experimental result, which we submit is of an importance amply sufficient to merit publication in *Nature*.

Sincerely,

[Redacted]

[Redacted]

(on behalf of all co-authors)

Reviewer Reports on the First Revision:

Referee #1 (Remarks to the Author):

The authors did an excellent job of responding to referee comments. I have no further issues with this manuscript.

Referee #2 (Remarks to the Author):

The modifications to the manuscript are very good, and they improved this already excellent paper. However, it still does not quite meet the bar for Nature in terms of broad impact. All three referees seemed to agree on the reason for this in the first round, and though the authors imply that this was due to a misunderstanding of what was being presented, I don't think that's the case. While it's true that the claims being made in the manuscript are now more accurately represented, the actual significance of the results were clear the first time around. Thus, the manuscript is still not sufficiently broadly impactful for publication in Nature. It is of high significance to experts in the field because it represents an important experimental step toward addressing the question of maximum repetition rate for plasma accelerators, but it does not do enough on its own to address this question in order to warrant great interest from the scientific community at large.

Referee #3 (Remarks to the Author):

Following the review process and subsequent revisions, the manuscript is now very much improved. All of my initial concerns have been addressed. I now support the authors in that the experimental results and the manuscript findings should be viewed as an upper limit on the inter-bunch repetition rate. This is an important result for the field of PWFA but it might be more suitable for a specialized journal. Overall, the authors have addressed my concerns.